# Inhibition of Peroxidation Potential and Protein Oxidative Damage by Metal Mangiferin Complexes

**Alberto J. Nuñez-Selles [1,*], Lauro Nuevas-Paz [1] and Gregorio Martínez-Sánchez [2]**

[1]   Research Division, Universidad Nacional Evangélica (UNEV), Paseo de los Periodistas 54, Ensanche Miraflores, Distrito Nacional, Santo Domingo CP 10203, Dominican Republic; nuevaspaz@yahoo.es
[2]   Independent Researcher, Via Dalmazia 16A, 60126 Ancona, Italy; gregorcuba@yahoo.it
*   Correspondence: anunez@unev.edu.do; Tel.: +1809-481-6256

**Featured Application: Metal–mangiferin complexes as antioxidant and protecting agents against protein oxidative damage in adjuvant therapies of pathologies of high morbidity and mortality: neurodegenerative diseases, cancer, diabetes, and cardiovascular disorders.**

**Abstract:** Background: Metal coordination complexes of polyphenolic compounds have been claimed to have better antioxidant and protection against protein oxidative damage effects than the isolated ligands. Whereas flavonoids have been extensively studied, xanthones such as mangiferin are lacking extensive research. Methods: Cu (II), Zn (II), and Se (IV) mangiferin complexes were synthesized with different stoichiometric ratios. Products were isolated by preparative chromatography and subjected to spectral analysis by FT-IR, HPLC-DAD, and HPLC-ESI-MS. The inhibition effects on peroxidation potential and protein oxidative damage were determined for all the metal–MF complexes. Results: Eight metal–MF complexes were isolated. Cu (II)–MF complexes did not improve MF antioxidant/protective effects; Zn (II) complexes (stoichiometric ratio 1:2) antioxidant/protective effects had no significant differences to MF; Zn (II)– and Se (IV)–MF complexes (stoichiometric ratio 1:3) showed the best inhibition effects on peroxidation potential (49.06% and 32.08%, respectively), and on the protection against protein oxidative damage (14.49% and 20.81%, respectively). Conclusions: The antioxidant/protective effects of Se (IV)– and Zn (II)–MF coordination complexes were significantly improved as compared to isolated MF, when the reaction between the metal salt and MF was performed with a stoichiometric ratio 1:3.

**Keywords:** oxidative stress; protein oxidative damage; mangiferin; metal–mangiferin complexes; antioxidant therapy

## 1. Introduction

Studies about the possible therapeutical application of coordination complexes with bioactive organic ligands have demonstrated exceptional development in the first two decades of the 21st Century. Transition metals seems to be particularly interesting because of the broad spectrum of complex geometries, coordination numbers, and redox states as compared to other metal groups. Redox activities of these complexes and their influence on the homeostasis at the cellular level have been intensively investigated because of their antioxidant and protection against protein oxidative damage properties [1]. The first systematic study about the use of flavonoids as bioactive ligands for the synthesis of metal complexes was performed on 4′,7,8-trihydroxy-isoflavone with zinc, copper, manganese, nickel, cobalt, and selenium, and all the metal complexes had a higher antioxidant effect than the isolated isoflavone [2]. From all studied flavonoids, quercetin was shown to form the most stable metal complexes, having a significant higher antioxidant activity than the isolated flavonoid [3], and this has been demonstrated for Mn and Zn [4], Va [5], Co [6], Cu [7], and Sn [8]. Catechin and epicatechin metal complexes have demonstrated the capacity to protect DNA from oxidative damage [9], but these complexes have

a low stability [10]. However, xanthones like mangiferin (2-β-D-glucopyranosyl-1,3,6,7-tetrahydroxy-9H-xanthen-9-one) have not been studied extensively. Mangiferin (MF) has been shown to be not only a potent antioxidant but also a potential candidate for the treatment of neurodegenerative diseases [11]. Its cardiovascular, liver, and brain protecting activities have been related to its capacity to protect cells from oxidative damage [12]. We reported previously that MF had a protective effect against protein oxidative damage in serum, liver, and brain from mice [13]. However, its poor water solubility has always brought the question of how to improve MF bioavailability, and several procedures have been attempted for this goal, including its integration to polymer systems for controlled release formulations [14–16]. However, there is the risk to reduce MF antioxidant properties, i.e., when it has been nano-encapsulated with β-lactoglobulin to increase its oral bioavailability [17]. The formation of MF metal complexes in plant aqueous extracts seems to be a possibility to explain that sometimes the biological activity of the MF-containing extract is higher than the isolated MF [11]. We investigated the presence of copper (II), zinc (II), and selenium (IV) in several varieties of mango trees and found that Se concentration was consistent among the varieties while Cu and Zn concentrations varied according to the mango variety [18]. Therefore, we hypothesized that the combination of MF with those micronutrients (Cu, Zn, and/or Se), probably through MF metal complexes, was a possible explanation for such behavior. The alternative to enhance MF antioxidant mechanism through the formation of metal complexes appears to be a possible route to increase not only its bioavailability but bioactivity. We report the synthesis, spectral characterization, and evaluation of the peroxidation potential and protection against protein oxidative damage of MF metal complexes with Cu (II), Zn (II), and Se (IV) for the first time to demonstrate the possibility to develop new compounds with the capability to increase the antioxidant effect and protection against protein oxidative damage as compared to the isolated ligand (MF). New metal–MF complexes might be useful in antioxidant adjuvant therapies for neurodegenerative diseases treatment such as cancer, diabetes, and cardiovascular disorders among others.

## 2. Materials and Methods

### 2.1. Chemicals

MF (95%) was purchased from N&R Industries Inc. (Shaanxi, China); sodium selenite anhydrous (99%) from Thermo Sci, MA, USA; zinc (II) sulfate heptahydrate (99%); copper (II) sulfate pentahydrate (99%) and trichloroacetic acid (99.5%) from Macron, Mexico; and methanol (HPLC grade), ethanol (PA grade), ethyl acetate (PA grade), acetonitrile (HPLC grade), formic acid (88%), hydrochloric acid (35%), glacial acetic acid (99%), and sodium phosphate dibasic (99%) from JT Baker, USA. Sodium phosphate monobasic (99%) and potassium iodide (99.5%) were purchased from Fisher Chemical (Fair Lawn, NJ, USA). N-Methyl-2-phenyl-indole (99%), malonyldialdehyde (99%), human haemoglobin, 2,4-dinitro-phenylhydrazine (99%), trichloroacetic acid (88%), guanidine hydrochloride (99%), and silica gel G-25 were purchased from Sigma-Merck, St. Louis, MO, USA.

### 2.2. Synthesis of Metal-Mangiferin Complexes

Pure MF (95%, HPLC), 42.2 g, was dissolved in 900 mL HPLC grade methanol in a 1 L volumetric flask, the solution was adjusted to pH 6.5 by addition of 0.1 M solution of hydrochloric acid, and the volume was filled to yield 0.1 M MF solution for chemical synthesis. The corresponding amount of metal salt was dissolved in distilled water in a 1 L volumetric flask, and the solution (0.1 M) was adjusted to pH 6.5 in the same way. Metal salt solution (100 mL) was poured into a 1 L two-necked flat-bottomed flask provided with a water bath at 30 °C and an electromagnetic stirrer. A thermometer was placed in one neck and a dropping funnel in the second neck to add the MF solution dropwise (200 mL or 300 mL for stoichiometry 1:2 or 1:3, respectively) during 10 min. The solution was stirred for 2 h and the yellowish precipitate was filtered through sintered

glass, washed thoroughly with methanol, and poured into a desiccator overnight. Each batch was replicated three times.

### 2.3. Preparative Chromatography

The metal–MF complexes were fractionated by preparative column chromatography with a glass column (20 × 5 cm) packed with silica gel G-25. The reaction mixtures were eluted with methanol acidified with acetic acid (pH = 6.5) at 5 mL/min. Collected fractions were grouped according to retention times and the solvent was evaporated under vacuum. Purified metal–MF complexes were left in a desiccator overnight, weighed, and packed in 2 mL Eppendorf vials for subsequent analyses.

### 2.4. HPLC-DAD Analysis of Purified Metal-MF Complexes

HPLC-DAD analyses of purified metal–MF complexes were performed with a Young Lin HPLC System (South Korea) equipped with a YL-9110 quaternary pump, YL-9150 autosampler (fitted with a 20 μL loop), and a YL-9160 diode-array detector (DAD) coupled to a data acquisition and processing system (Clarity software). A column (RP-18, 5 μm, 250 × 4 mm i.d., Merck, Germany) was placed in a YL-9131 column oven at 30 °C. Solvents were degassed (YL-9101), and injection volume was 20 μL. Analyses were carried out by gradient elution with two solvents (A = acetic acid (0.1%) in water; B = acetic acid (0.1%) in methanol). The ratio of A:B was increasing from 9:1 to 1:9 in 35 min at a flow rate of 1 mL/min. Detection wavelength was fixed at 278 nm.

### 2.5. Sample Preparation

For scanning electron microscopy-energy dispersive spectroscopy (SEM-EDS) analyses, samples were mounted onto double-sided carbon tape previously adhered to aluminum stubs. Analyses were performed in low vacuum at 20 kV. For Fourier-transformed infra-red spectroscopy (FT-IR) analyses, samples were mounted onto two diamond micro cells, compressed to obtain a thin film, and analyzed using a Continuum IR microscope. For high performance-electron spray interface-mass spectrometric (HPLC-ESI-MS) analyses and preparative chromatography experiments, samples (50 mg) were dissolved in methanol (25 mL).

For biological evaluations, blood samples (200 μL), were collected by a finger puncture from healthy subjects with a sterilized lancet and transferred into an Eppendorf vial containing 2 mL saline, and then centrifuged at 3000× *g*. The supernatant was discarded, and 2 mL of double-distilled water was added to the erythrocyte fraction to induce an osmotic shock and the consequence lysis. Fractions of erythrocyte lysate (RBC, 200 μL) were collected with a micropipette, transferred into Eppendorf vials (1 mL), coded, and maintained at −10 °C in a vertical freezer until processing within the next 30 days.

Incubation was performed using disposable Petri dishes (Fisherbrand, Pittsburgh, USA) in a DNI-150 incubator (MRC, Israel); samples were centrifuged using a MiniSpin centrifuge (Eppendorf, Germany); blood samples were collected with AutoLancets (Palcolabs, Santa Cruz, CA, USA) in 0.5, 1, and 2 mL-PVC vials (Eppendorf, Germany). Solutions were prepared with bidistilled water (Aqua MAXTM-Ultra 372, Seoul, Korea).

### 2.6. Equipment

Melting points were measured with a MPA-12 equipment (MRC, Israel). SEM-EDS analyses were performed on a JEOL 6480 (Japan) LV Scanning Electron Microscope equipped with an EDAX, X-ray Fluorescence Detecting Unit. The EDS system was used in combination with the SEM system to obtain the elemental composition of the samples. FT-IR analyses were performed on a Thermo iS50 equipped with a Continuum IR microscope (Thermo Sci, USA). Mass spectra were recorded with a Thermo Finnigan LCQIon Trap (Thermo Separation, USA) with electrospray ionization on negative mode between 50 and 1500 Da; collision chamber temperature 375 °C; pressure 4.1 bar; dry nitrogen flow as nebulizer, 10 mL/min, and helium as collision gas. First and second order fragmenta-

tion patterns were recorded at 1.2 and 1.5 V, respectively. Chromatographic analysis of reaction mixtures was performed on a Waters 600E-HPLC (CA, USA) with a RP-18 column, 250 × 4.5 mm (Kromasil, Sweden) at 30 °C. Injection volume was 5 μL. Separation was carried out by gradient elution with two solvents (A = formic acid (0.1%) in water; B = formic acid (0.1%) in acetonitrile). The ratio of A:B was increasing from 9:1 to 10:0 in 90 min at a flow rate of 1 mL/min. Data acquisition and peak integration analysis was performed using XCalibur software. Preparative chromatography was performed on a CF-7 fraction collector (Spectrum Labs, Canada) with 1 min for each collecting tube (5 mL). The eluates were analyzed by a YL-9160 diode-array detector (DAD) coupled to a data acquisition and processing system (Clarity software). Eluates were dried on a R100 vacuum rotary evaporator (MRC, Israel). Absorbance values for antioxidant evaluations were determined with a Genesis 10S spectrophotometer (Thermo Sci, MA, USA) in 1 cm quartz cells.

### 2.7. Inhibition of Peroxidation Potential

The inhibition of peroxidation potential was determined by a modified method [19]. RBC lysates were incubated with copper (II) sulfate, 2 mM, and an equivalent concentration 0.1 M for MF and metal–MF complexes at 37 °C for 24 h. Malondialdehyde (MDA) was determined at 0 and 24 h (at 586 nm) after incubation. A control value was determined without adding samples. A calibration curve using standard MDA was constructed in the range 0–50 μM. The difference in MDA values was assumed as the peroxidation potential (PP), and its inhibition was calculated as a percentage against the control sample (for MF) or MF (for metal–MF complexes)

### 2.8. Inhibition of Protein Degradation

The inhibition of protein degradation was determined through the determination of carbonyl groups (CO) generated by the rupture of the protein peptidic bond [20]. RBC lysates were suspended in 1 mL hydrochloric acid (2 M), 1 mL 2,4-dinitrophenylhydrazine (0.2%), and an equivalent concentration 0.1 M for MF or MF–metal complexes. The solutions were incubated with stirring at 37 °C for 1 h. Then, 1 mL 10% trichloroacetic acid (TCA) was added, and the precipitate was extracted with 10 mL ethanol:ethyl acetate (1:1). The solid was reprecipitated with TCA (10%). Protein control sample (egg albumin) was dissolved in 1 mL guanidine hydrochloride in potassium phosphate buffer (0.1 M, pH = 6.5). The samples and the control were centrifuged, and absorbance was determined at 370–375 nm. The inhibition of CO production was calculated as a percentage against the control sample (for MF) or MF (for metal–MF complexes).

### 2.9. Ethical Approval

The protocol for human subjects was approved by the Bioethics Commission, National Evangelic University (UNEV), and submitted to the National Commission of Health Bioethics (CONABIOS) of the Dominican Republic. Final approval was registered as document 008-2016 [21]. Healthy volunteers (30) were recruited with their informed consent according to the Declaration of Helsinki from May–August 2016. Research assistants explained the purpose of the study to the volunteers, and the informed consent form was signed in the presence of a witness. All information about subject identities was kept confidential, and the corresponding codes were used to identify the samples.

### 2.10. Data Processing

All samples were analyzed by triplicate in every assay, and the results were expressed as $(\bar{Y}) \pm$ SD. Data from all experiments were analyzed using SpSS 9.0 software. The non-parametric Friedman test was used to detect changes within samples using Wilconxon's paired test. The Mann–Whitney U test was used to estimate significant differences ($p < 0.05$) between samples. The results of biological evaluations were expressed as the mean $\pm$ standard error of the mean (SEM) using a percentage scale against the control values for determining the inhibition capacity of MF and metal–MF complexes.



## 3. Results

### 3.1. HPLC-ESI-MS Analysis of Reaction Mixtures

HPLC-ESI-MS analytical results of reaction products are shown in Table 1. The reaction of MF with Cu (II) and Zn (II) sulfates, with the same stoichiometric ratios (1:2), rendered a mixture of two different types of complexes at retention times (RT), 25.8 and 27.5 min and 26.2 and 28.8 min, respectively. However, the reaction of MF with sodium selenite, at the same ratio (1:2), rendered only one peak at RT 33.5 min. The reaction of MF with Zn (II) sulfate (ratio 1:3) rendered a mixture of three different types of complexes with RT 26.4, 28.6, and 41.4 min. The reaction of MF with sodium selenite (ratio 1:3) rendered two peaks at RT 33.3 and 44.5 min. The reaction of MF with Cu (II) sulfate (ratio 1:3) rendered the same results (two peaks) as with ratio 1:2 with similar RTs. Molecular ions [M − H]⁻ of all chromatographic peaks indicated that three different types of complexes were obtained.

**Table 1.** Chromatographic retention times and molecular ions [M − H]⁻ of mangiferin and metal–mangiferin complexes by HPLC-ESI-MS.

| Sample | Peak 1 | | Peak 2 | | Peak 3 | | Peak 4 | |
|---|---|---|---|---|---|---|---|---|
| | RT (min) | [M − H]⁻ | RT (min) | [M − H]⁻ | RT (min) | [M − H]⁻ | RT (min) | [M − H]⁻ |
| MF | 17.8 ± 0.2 | 421 | - | - | - | - | - | - |
| Cu (1:2) | 17.6 ± 0.2 | 421 | 25.8 ± 0.4 | 905 | 27.5 ± 0.4 | 903 | - | - |
| Zn (1:2) | 17.8 ± 0.3 | 421 | 26.2 ± 0.2 | 907 | 28.8 ± 0.4 | 905 | - | - |
| Zn (1:3) | 17.4 ± 0.4 | 421 | 26.4 ± 0.3 | 907 | 28.6 ± 0.4 | 905 | 41.4 ± 0.6 | 1388 |
| Se (1:2) | 17.6 ± 0.4 | 421 | - | - | 33.5 ± 0.5 | 934 | - | - |
| Se (1:3) | 18.0 ± 0.4 | 421 | - | - | 33.3 ± 0.5 | 934 | 44.5 ± 0.5 | 1447 |

Legend: Sample describes the salt metal ion and the number between brackets mean stoichiometric ratio metal salt:mangiferin for the reaction; RT: retention time; [M − H]⁻: molecular ion in negative mode mass spectrometry (*m/z*); MF: mangiferin.

Fragmentation patterns (MS1 and MS2) showed the typical fragmentation peaks of MF at *m/z* = 421, 331, 271, and 259 for all peaks, indicating that metal–MF complexes were formed. [M − H]⁻ values of high-resolution mass spectra for Peak 2 were *m/z* = 906.19 for Cu (II) and *m/z* = 908.03 for Zn (II). Markedly, the reaction of MF with sodium selenite did not form a complex of such structure, probably because the orbital configuration of the Se atom. The isotopic peak clusters for the [M − H]⁻ ions at *m/z* = 906.19 (69% relative abundance) and *m/z* = 908.19 (31%) for Cu (II)–MF complex, and *m/z* = 908.03 (48%), 910.05 (28%), and 912.05 (19%) for Zn (II)–MF complex, indicated the presence of these metal–MF complexes (Figure 1A,B) [22]. [M − H]⁻ values of high-resolution mass spectra for peak 3 were *m/z* = 904.18 for Cu (II), *m/z* = 906.01 for Zn (II), and *m/z* = 937.60 for Se (IV). The isotopic peak clusters for the [M − H]⁻ ions of Cu (II)–MF and Zn (II)–MF complexes with their relative abundances, as described before, indicated the metal coordination (Cu (II) or Zn (II)) to the ligand. In the case of Se (IV)–MF complex, the isotopic peak cluster for the [M-H] ions were more complex (five peaks). [M − H]⁻ values of high-resolution mass spectra for Peak 4 were *m/z* = 1389.68 for Zn (II), and *m/z* = 1448.84 for Se (IV). The isotopic peak clusters for the [M − H]⁻ ions of Zn (II)–MF as described before. The isotopic peak cluster for the of Se (IV)–MF complex [M − H]⁻ ion (peak 4) at *m/z* = 1448.84 (50%), 1446.71 (24%), 1444.60 (9%), 1450.66 (9%), and 1445.62 (8%) indicated the presence of the Se (IV)–MF complex (Figure 1C).

### 3.2. Preparative Chromatography of Metal-MF Complexes

As per Table 1, compounds corresponding to Peaks 2 and 3 (ratio 1:2) for the reactions of Cu (II) and Zn (II) sulfates yielded 164 mg (Cu-1) and 186 mg (Cu-2), and 221 mg (Zn-1) and 234 mg (Zn-2), respectively, of yellowish crystalline powders with m.p. > 300 °C decomposed after preparative column chromatography. The compound corresponding to

peak 3 (Se-1) for the reaction of sodium selenite with MF (ratio 1:2) yielded 338 mg of a yellowish crystalline powder with m.p > 300 °C decomposed. Compounds corresponding to Peaks 2, 3, and 4 for the reaction of Zn (II) sulfate with MF (ratio 1:3) yielded 110 mg (Zn-3), 105 mg (Zn-4), and 198 mg (Zn-5), respectively. Compounds corresponding to Peaks 3 and 4 for the reaction of sodium selenite with MF (ratio 1:3) yielded 248 mg (Se-2) and 367 mg (Se-3), respectively, of crystalline or amorphous yellowish powders, respectively, with m.p. >300 °C decomposed. The purities of all compounds (10) were assessed by HPLC-DAD (>90% purity) with non-reacted MF as the main impurity (Peak 1 in all chromatograms). UV spectra for all metal–MF complexes showed the four characteristic absorption maxima of MF ($\lambda_{max}$ = 240, 255, 320, and 365 nm) but no significant differences were observed regarded to MF as reference.

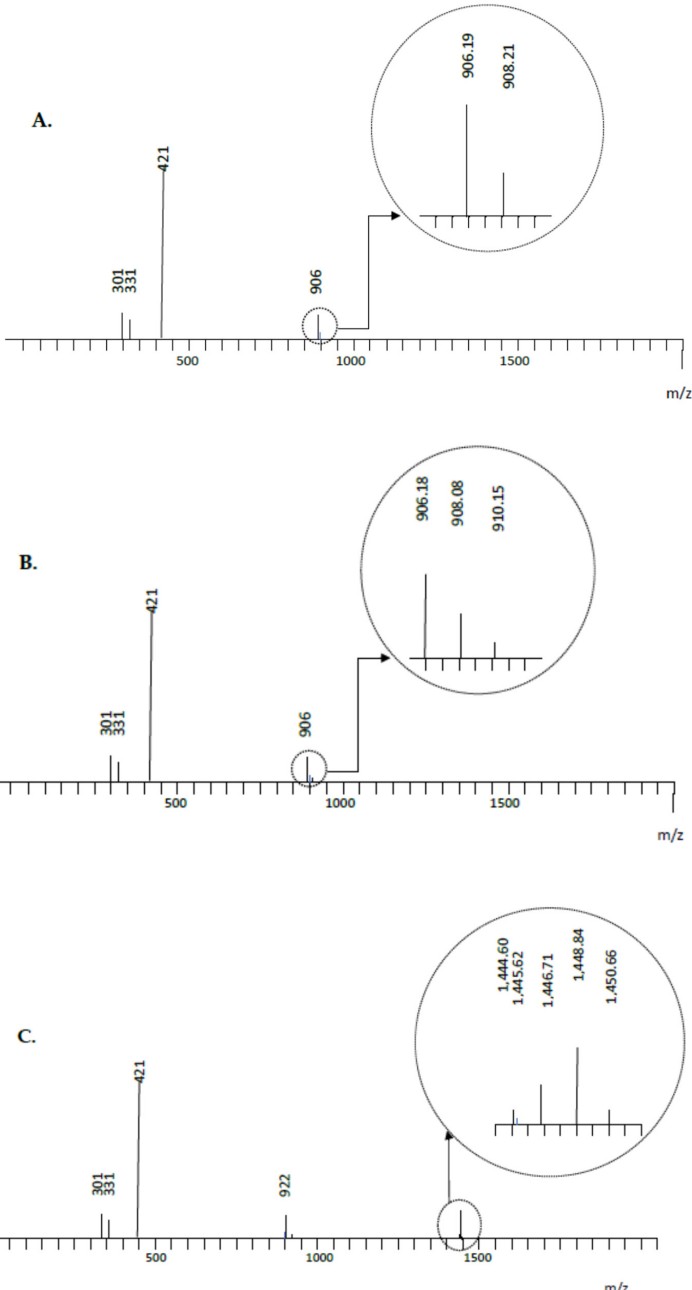

**Figure 1.** (**A**). Isotopic peak cluster of Cu (II)–MF complex [M − H]$^-$ ion, Peak 2. (**B**). Isotopic peak cluster of Zn (II)–MF complex [M − H]$^-$ ion, Peak 3. (**C**). Isotopic peak cluster of Se (IV)–MF complex [M − H]$^-$ ion, Peak 4.

### 3.3. Elemental Analysis of Metal-Mangiferin Complexes

Table 2 shows the results of SEM-EDS analyses of purified metal–MF complexes and its comparison to MF as a reference. The molecular formulae of metal–MF complexes were assessed according to the theoretical atomic composition and the values of $[M - H]^-$ ions recorded by high-resolution mass spectrometry.

**Table 2.** Elemental analysis data of mangiferin and purified metal–mangiferin complexes by scanning electron microscopy–energy dispersive spectroscopy (SEM–EDS).

| Sample | RT (min) | $[M - H]^-$ (m/z) | Elemental Composition | | | | | Others |
|--------|----------|-------------------|-------|-------|------|------|------|--------|
| | | | C | O | Cu | Zn | Se | |
| MF | 17.8 ± 0.2 | 421 | 55.61 | 40.66 | | | | 0.34 |
| Cu 1 | 25.8 ± 0.4 | 905 | 48.41 | 36.20 | 0.06 | | | 0.11 |
| Cu 2 | 27.5 ± 0.4 | 903 | 49.01 | 37,14 | 0.07 | | | 1.49 |
| Zn 1 | 26.2 ± 0.2 | 907 | 48.51 | 37.20 | | 0.08 | | 1.24 |
| Zn 2 | 28.8 ± 0.4 | 905 | 48.42 | 37.31 | | 0.07 | | 1.00 |
| Zn 3 | 26.4 ± 0.3 | 907 | 48.48 | 37.18 | | 0.08 | | 1.07 |
| Zn 4 | 28.6 ± 0.4 | 905 | 48.43 | 37.33 | | 0.07 | | 1.03 |
| Zn 5 | 41.4 ± 0.6 | 1388 | 46.88 | 35.97 | | 0.09 | | 2.14 |
| Se 1 | 33.5 ± 0.5 | 934 | 48.88 | 39.51 | | | 0.08 | 3.12 |
| Se 2 | 33.3 ± 0.5 | 934 | 48.70 | 39.47 | | | 0.08 | 2.96 |
| Se 3 | 44.5 ± 0.5 | 1447 | 47.13 | 38.54 | | | 0.11 | 3.22 |

Legend: RT: Retention time; $[M - H]^-$: molecular ion in negative mode mass spectrometry (m/z); MF: mangiferin. Sample nomenclature according to preparative chromatography fractions.

### 3.4. FT-IR Analyses of Metal–Mangiferin Complexes

For subsequent FT-IR analysis, samples were grouped according to similarities in RT, [M-H}⁻, and elemental composition, assuming that the same metal–MF complexes were formed using different stoichiometric ratios. Spectral signals of FT-IR of purified metal–MF complexes, according to this nomenclature, and its comparison to MF as a reference are shown in Table 3.

The appearance of the bands between 817 and 818 cm⁻¹ in all FT-IR spectra for metal–MF complexes (Table 3) was indicative of the metal–oxygen bond formation, thus the presence of coordination complex of MF as ligand with Cu (II), Zn (II), and Se (IV) as the metal nuclei. As an example, Figure 2 shows the superposed FT-IR spectra of Se (IV)–MF complexes with different stoichiometric ratios, where bands at 817.6 and 817.8 cm⁻¹ appeared for Peaks 3 and 4, respectively. Similar FT-IR spectra were obtained for all metal–MF complexes as regarded the 800–1600 cm⁻¹ region.

### 3.5. Inhibition of Peroxidation Potential and Protection against Protein Oxidative Damage

Table 4 shows the data of the metal–MF complexes effects on the peroxidation potential through the MDA differential lectures and the protection against protein oxidative damage through the determination of CO groups in RBC lysates. Whereas both Zn (II), Complexes I and II, and Se (IV), Complexes II and III, decreased the peroxidation potential, Cu (II) complexes did not influence MDA concentration as compared to MF. Interestingly, Cu (II)–MF complexes showed higher values of MDA concentration than MF. Se (IV), Complex III, had the highest significant inhibition percentage of peroxidation potential as compared to the rest of synthesized metal–MF complexes (see Figure 3), but also it was the most potent inhibitor of protein degradation. Cu (II) and Zn (II)–MF complexes did not improve the protection capacity of MF against protein degradation.

**Table 3.** Spectral analysis data of FT-IR spectra of mangiferin and metal–mangiferin complexes.

| Signal | Sample | | | | |
|---|---|---|---|---|---|
| | MF | Metal/Ratio | Peak 2 | Peak 3 | Peak 4 |
| O-H$_{str}$ | 3366.31 | Cu (1:2) | 3365.69 | 3366.14 | |
| | | Zn (1:2) | 3366.57 | 3365.57 | |
| | | Zn (1:3) | 3366.57 | 3365.57 | 3366.76 |
| | | Se (1:2) | | 3366.70 | |
| | | Se (1:3) | | 3366.70 | 3366.85 |
| C-H$_{str}$ | 2971.68 | Cu (1:2) | 2940.07 | 2939.76 | |
| | | Zn (1:2) | 2915.87 | 2939.53 | |
| | | Zn (1:3) | 2915.87 | 2939.53 | 2943.07 |
| | | Se (1:2) | | 2939.20 | |
| | | Se (1:3) | | 2939.20 | 2939.34 |
| C=O$_{str}$ | 1648.39 | Cu (1:2) | 1648.62 | 1650.09 | |
| | | Zn (1:2) | 1649.13 | 1651.31 | |
| | | Zn (1:3) | 1649.13 | 1651.31 | 1651.34 |
| | | Se (1:2) | | 1648.37 | |
| | | Se (1:3) | | 1648.37 | 1648.87 |
| CH-CH$_{str}$ | 1489.16 | Cu (1:2) | 1488.16 | 1495.18 | |
| | | Zn (1:2) | 1487.94 | 1493.27 | |
| | | Zn (1:3) | 1487.94 | 1493.27 | 1488.34 |
| | | Se (1:2) | | 1487.16 | |
| | | Se (1:3) | | 1487.16 | 1490.93 |
| C-O$_{str}$ | 1255.37 | Cu (1:2) | 1255.86 | 1254.83 | |
| | | Zn (1:2) | 1255.51 | 1254.81 | |
| | | Zn (1:3) | 1255.51 | 1254.81 | 1255.86 |
| | | Se (1:2) | | 1294.58 | |
| | | Se (1:3) | | 1294.58 | 1254.86 |
| C-C$_{str}$ | 1091.14 | Cu (1:2) | 1095.84 | 1095.94 | |
| | | Zn (1:2) | 1095.35 | 1095.58 | |
| | | Zn (1:3) | 1095.35 | 1095.58 | 1095.19 |
| | | Se (1:2) | | 1076.47 | |
| | | Se (1:3) | | 1076.47 | 1095.49 |
| M-O$_{str}$ | NA | Cu (1:2) | 817.8 | 818.1 | |
| | | Zn (1:2) | 818.3 | 818.6 | |
| | | Zn (1:3) | 817.8 | 818.1 | 817.8 |
| | | Se (1:2) | | 818.1 | |
| | | Se (1:3) | | 817.6 | 817.8 |

Legend: MF: Mangiferin, Metal/Ratio: Metal from salts employed in reactions/Stoichiometric molar ratio of reactants (metal salt:MF), Peaks: Chromatographic peaks as per Table 1.

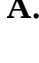

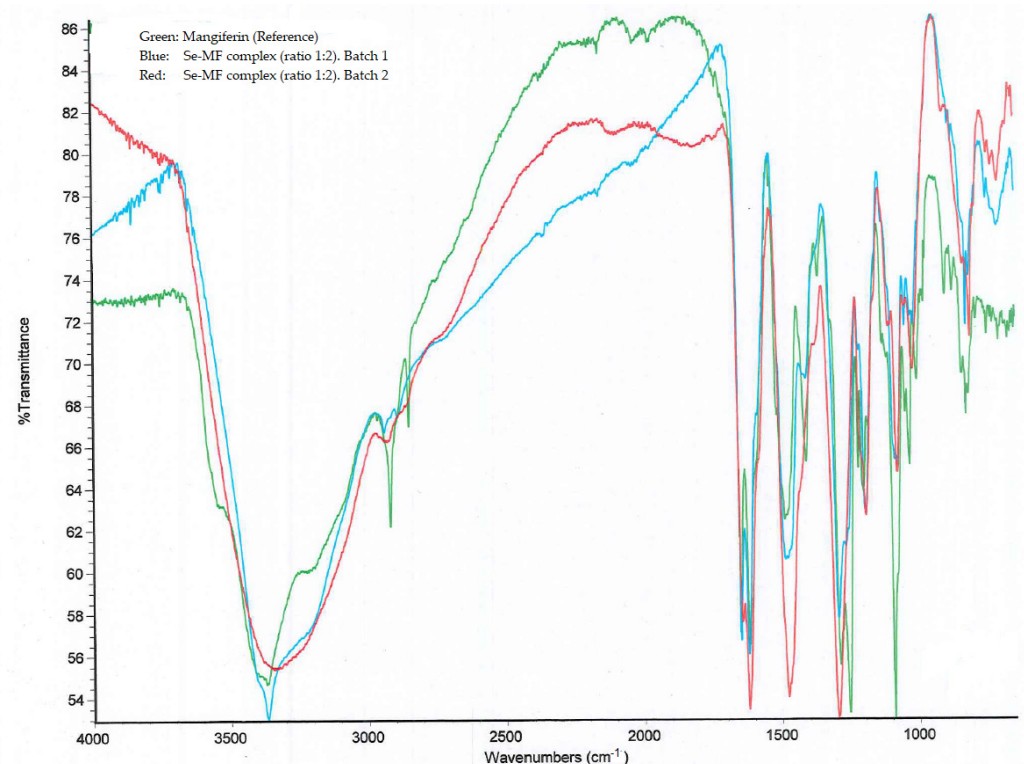

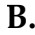

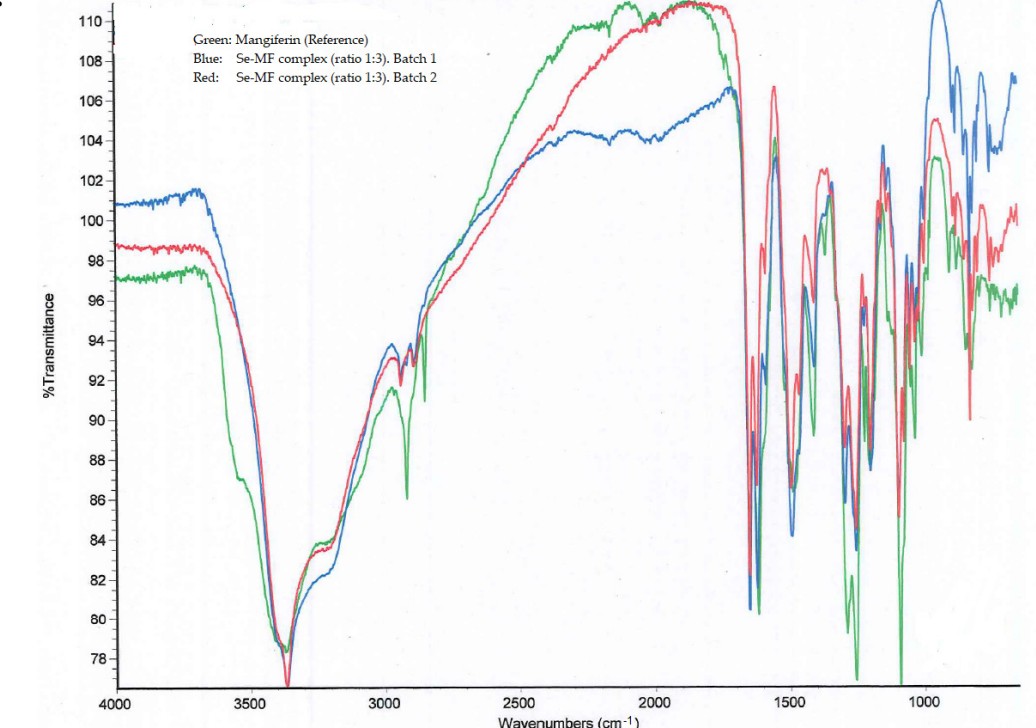

**Figure 2.** FT-IR spectra of mangiferin and selenium (IV)-mangiferin complexes in the region between 800 and 1800 cm$^{-1}$. The appearance of bands at 817.6 cm$^{-1}$ for Peak 3, ratio 1:2 (**A**) and 817.8 cm$^{-1}$ for Peak 4, ratio 1:3 (**B**) indicated the formation of metal–oxygen bonds of Se (IV)-mangiferin complexes. The same occurred for all metal–mangiferin complexes described herein.

**Table 4.** Determination of the inhibition of the peroxidation potential and the protein degradation by metal–mangiferin complexes.

| Sample | Chromatogr. Peak | PP (μM) | CO (μM) | Inhibition (%) | |
| | | | | PP | Protein Degradation |
|---|---|---|---|---|---|
| Control | | 16.4 ± 1.1 [a] | 504.8 ± 50.5 [a] | - | - |
| Mangiferin | 1 | 15.9 ± 1.5 [a] | 490.6 ± 55.2 [a] | 3.10 | 2.81 |
| Cu (1:2) | 2 | 16.0 ± 1.4 [a] | 497.0 ± 48.4 [a] | 0.01 | NA |
| | 3 | 16.2 ± 1.6 [a] | 501.4 ± 49.7 [a] | 0.02 | NA |
| Zn (1:2) | 2 | 13.8 ± 1.5 [ab] | 456.8 ± 50.5 [a] | 13.21 | 6.94 |
| | 3 | 14.4 ± 1.5 [ab] | 472.9 ± 45.8 [a] | 9.43 | 3.67 |
| Zn (1:3) | 4 | 10.8 ± 1.2 [abc] | 419.0 ± 56.8 [ab] | 32.08 | 14.49 |
| Se (1:2) | 3 | 12.6±1.4 [abc] | 444.9±52.2 [ab] | 20.75 | 12.24 |
| Se (1:3) | 4 | 8.1±1.7 [bcd] | 388.0±55.0 [ab] | 49.06 | 20.81 |

RBC lysates were tested adding equivalent amounts of 20 mM for each tested compound. PP represents MDA values according to differential lectures at 0 and 24 h (at 586 nm) after incubation of RBC lysates with tested compounds. CO values according to absorbance lectures at 370–375 nm. Inhibition values as percentages against the control sample (for mangiferin) or mangiferin (complexes). Legend: Control, egg albumin; PP, peroxidation potential; CO, carbonyl groups. NA, Not Applied. Different letters mean significant variation ($p > 0.05$).

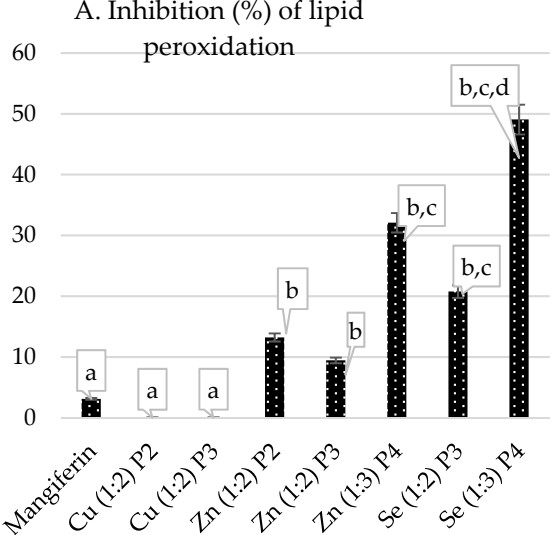

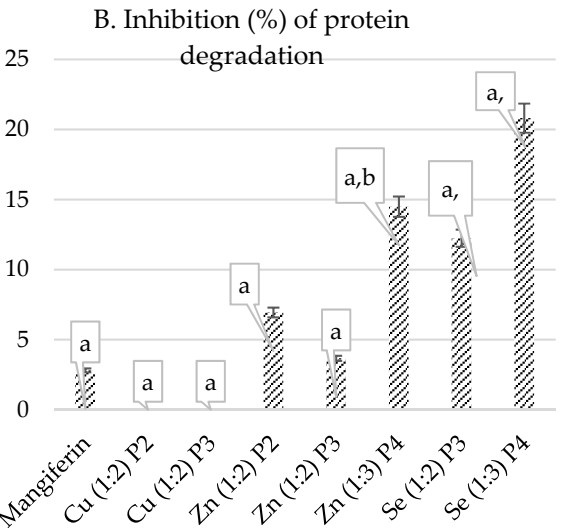

**Figure 3.** Determination of the inhibition of the peroxidation potential and the protein degradation by metal–mangiferin complexes. (**A**). Percentage of inhibition of lipid peroxidation; (**B**). Percentage of inhibition of protein degradation. Inhibition values as percentages against the control sample (for mangiferin) or mangiferin (complexes). Legend: Cu (1:2) P2, copper complex 1:2 Chromatographic peak 2; Cu (1:2) P3, copper complex 1:2 Chromatographic peak 3; Zn (1:2) P2, zinc complex 1:2 Chromatographic peak 2; Zn (1:2) P3 zinc complex 1:2 Chromatographic peak 3; Zn (1:3) P4 zinc complex 1:3 Chromatographic peak 4; Se (1:2) P3, selenium complex 1:2 Chromatographic peak 3; Se (1:3) P4, selenium complex 1:3 Chromatographic peak 4. Different letters mean significant variation ($p > 0.05$).

## 4. Discussion

### 4.1. Metal–Mangiferin Complexes

One of the most complex problems in the synthesis of metal–organic complexes is the stoichiometric ratio that leads to the most stable complex. Said relationship will depend not only on the nature of the metal but also on the structure of the ligand. A 1:3 stoichiometric

ratio can give rise to complexes where several ligands may be attached to a metal nucleus, but a 1:2 ratio rendered a complex with two ligands attached to said nucleus [3]. It has been reported that the coordination sites can be two adjacent hydroxyl groups [2]. The metal–MF complex of Peak 2 is probably the result of the metal coordination to the quinolic oxygen (carbonyl group) and one adjacent hydroxyl group from two MF moieties, but that possible structure was only observed for Cu (II) and Zn (II) complexes (ratio 1:2). That Peak 2 was not observed for the reaction between sodium selenite and MF at the same ratio, probably due to a steric hindrance. These type of structures with Cu (II) and Fe (II), but with a different xanthone (β-mangostin), have been reported [23]. Similar Cu (II) and Zn (II) coordination complexes have been also reported, but with a different xanthone ligand (isoeuxanthone) [24]. Other metal–MF complexes with Cu (II) and Zn (II) have been reported, but only with one MF moiety attached to the metal nucleus [25], probably due to the basic reaction conditions (pH = 7.5), while our reactions were conducted in acidic medium (pH = 6.5). The metal–MF complexes of Peak 3 were observed for the reaction between Cu (II), Zn (II), and Se (IV) salts with MF. The coordination probably occurred through two adjacent hydroxyl groups from two different MF moieties as it has been reported for ruthenium (II) coordination complex with MF [26]. That Peak 3 was not observed for the reaction between Cu (II) sulfate and MF at the same ratio. Structures of this type but with Ge (IV) coordinated to flavonoid polyphyenols have been reported [27], but not with MF. SEM-EDS and ESI-MS data corroborated that Cu (II) and Zn (II) complexes (Peaks 2 and 3) were dihydrated, whereas Zn (II)–MF (peak 4) was tetrahydrated. All Se (IV) complexes were anhydrous. According to these results, molecular formulae of metal–MF complexes were estimated as shown in Table 5.

**Table 5.** Molecular formulae and molecular masses of metal–MF complexes.

| Metal Nucleus | Peak | Stoichiometric Ratio | Predicted Complex Molecular Formula | Estimated Complex Molecular Mass |
|---|---|---|---|---|
| Copper (II) | 2 | 1:2 | $C_{38}H_{34}O_{22}Cu.2H_2O$ | 942.22 |
| | 3 | 1:2 | $C_{38}H_{32}O_{22}Cu.2H_2O$ | 940.21 |
| Zinc (II) | 2 | 1:2 | $C_{38}H_{34}O_{22}Zn.2H_2O$ | 944.06 |
| | 3 | 1:2 | $C_{38}H_{32}O_{22}Zn.2H_2O$ | 942.04 |
| | 4 | 1:3 | $C_{57}H_{46}O_{33}Zn_2.4H_2O$ | 1461.76 |
| Selenium (IV) | 3 | 1:2 | $C_{38}H_{34}O_{23}Se$ | 937.61 |
| | 4 | 1:3 | $C_{57}H_{46}O_{33}Se_2$ | 1448.85 |

Legend: Peak as per Table 1. Molecular formulae and masses were estimated as per Table 2.

Further experimental work is needed for complexes structures elucidation including [1]H- and [13]C-nuclear magnetic resonance techniques and single-crystal X-ray diffraction in order to correlate complex structure with its antioxidant and protective against protein oxidation effects.

*4.2. Evaluation of Antioxidant and Oxidative Damage Protection Properties of Metal-Mangiferin Complexes*

Adjuvant antioxidant therapies have been claimed to improve standard therapies for neurodegenerative diseases and cardiovascular disorders [28], but it seems that more important than the antioxidant effect, which means the reduction of ROS and/or the stimulation of the endogenous antioxidant defense mechanism, is the protection to oxidative damage thus maintaining the integrity of DNA within the organism [29]. Products having both antioxidant and protective effect against oxidative damage would be the best choice to those adjuvant therapies [30]. Antioxidants and pro-oxidants cannot be seen only in terms of cell damage/protection, but they play an important role in cell signal transduction pathways, both in physiological and pathological conditions [31]. For instance, the redox modulation

o HIF1α (hypoxia-inducible factor 1 alpha), named "pseudohypoxia" is frequently used by cancer cells to promote glycolytic metabolism to support biomass synthesis for cell growth and proliferation [32]. MF inhibits the expression of HIF-1α [33]; however, even when MF has shown promising chemotherapeutic and chemopreventive potential, its bioavailability is limited [34]. MF derivates as metal complexes, or other structural modification as metal complexes with glucoside–sulfate conjugates or phospholipid complexes, may increase the bioavailability of MF and thus its potential use for prophylactics and/or therapeutics.

All previous research work about antioxidant and/or protection against protein oxidative damage of metal–polyphenols complexes has demonstrated that such coordination complexes had higher antioxidant and/protective effects than the isolated ligand. However, it cannot be generally assumed either for all metal nucleus or all polyphenolic ligands. Our results demonstrated that Cu (II)–MF complexes had similar peroxidation potential to MF, but protein oxidative damage was increased as compared to MF. Biomolecules damage by Cu-mediated hydroxyl radical production may be acting as a drawback for these Cu (II)–MF coordination complexes [35]. Se (IV)–MF (peak 4) showed the best inhibition values for both peroxidation potential (49.06%) and protection against protein oxidative damage (20.81%), followed by the same complex type of Zn (II)–MF with 32.08% and 14.49%, respectively. Further structure–activity relationships with accurate crystallographic data for this complex type would be recommended. However, the inhibition of the peroxidation potential for both Zn (II)–MF complexes (Peaks 2 and 3) and the protection against protein oxidative damage were not significantly different as compared to MF. Selenium fulfils several functions on different metabolic pathways, and these are mainly connected to its antioxidant properties, as an essential part of some antioxidant enzymes activities [36]. The higher effect of Se (IV)–MF complexes as compared to the isolated MF may be related to the synthesis of selenoproteins, which play essential roles in several diseases including neurodegeneration, cardiovascular disorders, and cancer. Thus, organometallic Se (IV)–MF complexes may influence functions of selenoproteins that might be considered for prevention and therapeutic treatment of those disorders [37].

However, the choice of the most effective, non-toxic, Se compound represents a very complex issue. Several Se salts, both inorganic and organic are being studied, to be used as a nutritional supplement or pharmaceutical. The results regarding the comparison of inorganic and organic compounds are not fully consistent, but in general, organic Se compounds show higher biological activities and lower toxicities [38]. Thus, Se (IV)–MF complexes would be investigated subsequently as a potential source of novel health compounds.

Metal–flavonoid complexes design allows to obtain compounds with improved biological and physicochemical properties. In particular, generating an increase of the flavonoid antioxidant properties [39]. Coordination sites, polyphenol structure, metal ion type, and the stoichiometric ratio metal:polyphenol are important factors that influence the antioxidant activity of these coordination complexes. Therefore, the biological evaluation of these compounds is of paramount importance for its application [40]. In the present study, Se (IV)–MF and Zn (II)–MF complexes showed a higher antioxidant effect than the isolated MF in vitro, without having pro-oxidant effect. Our findings confirmed the hypothesis about the possible effect of other components on the observed antioxidant and protective effects of *Mangifera indica* L. aqueous extract, rather than considering that MF was mainly responsible of these biological effects.

## 5. Conclusions

The antioxidant and protection against protein oxidative damage effects of Se (IV)– and Zn (II)–MF coordination complexes were significantly improved when the reaction between the metal salt and MF was performed with a stoichiometric ratio 1:3. Cu (II)–MF complexes had similar effect to MF in the inhibition of peroxidation potential but increased protein damage. Se (IV)–MF complexes showed their improved capability, as compared to Cu (II)- and Zn (II)–MF complexes, to be considered for further product development. Further studies with accurate spectroscopic and crystallographic data are recommended

for the understanding of the inhibition effects on lipid peroxidation and protection against protein oxidative damage of these chemical entities.

## 6. Patents

Patent application P0194/2018 (Oficina Nacional de la Propiedad Industrial, Dominican Republic) is pending for approval.

**Author Contributions:** Conceptualization, methodology, chemical investigation, writing—review and editing, project administration, supervision, and funding acquisition: A.J.N.-S.; chemical investigation, writing—review, and formal analysis: L.N.-P.; biological investigation, formal analysis, and writing—review: G.M.-S. All authors have read and agreed to the published version of the manuscript.

**Funding:** This research was co-funded by FONDOCYT, grant number 2014-1D4-135, Ministry of Higher Education, Science, and Technology (MESCYT), Dominican Republic, and the Universidad Nacional Evangélica (UNEV), Santo Domingo, Dominican Republic.

**Institutional Review Board Statement:** The study with human subjects was conducted according to the guidelines of the Declaration of Helsinki, and approved by the Comisión Nacional de Bioética en Salud (CONABIOS), Dominican Republic, protocol code 008-2016, approved on 27.04.2016.

**Informed Consent Statement:** Informed consent was obtained from all subjects involved in the study. Written informed consent has been obtained from the patients to publish this paper.

**Data Availability Statement:** Experimental data are available on request at anunez@unev.edu.do.

**Acknowledgments:** Thanks to the Materials Characterization Center, University of Puerto Rico (MCC/UPR) for spectral data. Thanks to Deyrel Reynoso and Keyla Salce (UNEV, Dominican Republic) for accounting and administrative support, respectively. Thanks to Natalia Vega (UNEV, Dominican Republic) for experiments on chemical synthesis.

**Conflicts of Interest:** One of the authors (AJNS) has a patent application pending (P0194/2018, Dominican Republic). The funders had no role in the design of the study; in the collection, analyses, or interpretation of data; in the writing of the manuscript; or in the decision to publish the results.

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
