# Peer review of "Inhibition of Peroxidation Potential and Protein Oxidative Damage by Metal Mangiferin Complexes"

_applsci, doi:10.3390/app12042240_

Round 1
Reviewer 1 Report
AJ Nuñez Selles has an established expertise in extraction and pharmaceutical effects of mangeferin. In this article they described the synthesis of Cu/Zn and Se-complexes by mangiferin and their characterization by IR, NMR and MS in a first part, and the anti-oxidant/protective effects of the metal complexes compared to those of mangiferin, in the last part. I am not convinced by the structure proposed for the metal-complexes. However, this has little influence on the anti-oxidant/protective effects of these complexes. I think the first part should be improved before the publication of this article in Applied Sciences, because the subject may be of interest to the scientific community.
Problems or questions
Did the authors observed in FT-IR spectra the appearance of band between 400 and 800 cm-1 in the presence of metal, due to M-O vibration? In high resolution mass spectra MS-ESI technic, the appearance of a cluster of isotopic peaks of the corresponding metal ion isotopes should also confirm that metal ions have been successfully coordinated to the ligand.
What is the uncertainties of the measure (IR, NRM…)? Are the observed shift significant to affirm the formation of metal-complexe? For example, the C=O vibration for peak 2 shifts only by 0.3 cm-1. In 1H NRM, the shift of O-H is only 0.1 to 0.3ppm for peak 3. In 13C NRM, the chemical shift of carbonyl is 179.5 ppm for MF and 179.3 ppm for Cu complex whereas the authors affirmed that CU is coordinated to MF by the C=O.
Maybe the superpositions of FT-IR, NRM spectra of MF in the absence and in the presence of metal are more relevant than table with a lot of data (just for one metal and one complex).
Line 266 how MS can confirm a structure? It can confirm the stoichiometry but not the structure.
Some additional UV-visible absorption spectra should distinguish the different complex. Doesn’t the HPLC-DAD give these absorption spectra?
The coordination via the glucopyranosyl group is not usual, and the data are not convincing. For complex III (with Se and Zn), is it possible to have a coordination between the catechol part and the metal and between metal and the carbonyl-hydroxyl moieties for the second chelation?
Typographical errors
I suggested simplifying the table 3, or splitting it into several.
Superscript « 2 » instead of « 3 » in the list of authors
The name of cycles in Mangeferin structure (A and B) is missing fig. 1
Line 78-79, select sulfate or sulphate
Line 213, Peak 2 for Cu or Zn complexes has different retention times. Add the end of the sentence : “25.8 min, and 27.5 min, and 26.2 and 28.8 min, respectively”
In table 3, for 13C NRM, and for glucopyranosyl carbon, replace C’ instead of C.
For clarity, simplify the structure of glucopyranosyl group in Fig.1
Line 265,… NMR signal of carbonyl group to 178 ppm, in the table it is 179.3. The 178 ppm value is for peak 3. Is there a confusion here?

Author Response
Thanks for all your comments and recommendations that we have appreciated indeed. It contributed to improve the manuscript at large. Please find enclosed the answers and comments to your points (text in red color)
Reviewer 2 Report
The study's authors titled “Inhibition of Peroxidation Potential and Protein Oxidative Damage by Metal Mangiferin Complexes” report the synthesis, spectral characterization and the evaluation of the peroxidation potential and protection against protein oxidative damage of mangiferin metal complexes with Cu (II), Zn (II) and Se (IV). The manuscript is well written; however, before publication manuscript needs to be revised. The part of the manuscript devoted to the antioxidant effect of MF metal complexes is not big that’s why in my opinion the data from the experiments: Inhibition of Peroxidation Potential and Inhibition of Protein Degradation should be presented in different form then table (which didn’t shows the data really clear). Determination of the inhibition of the peroxidation potential and the protein degradation by metal-mangiferin complexes should be presented separately in the form of charts with bars, where clearly the data are compared to the control.

Author Response
Thanks indeed for your valuable comments and suggestions. Your review helped us to improve the manuscript. Please find enclosed our reply to your comments (text in red color)
Round 2
Reviewer 1 Report
2nd review
Inhibition of Peroxidation Potential and Protein Oxidative Damage by Metal Mangiferin Complexes by Alberto J. Nuñez Selles, Lauro Nuevas Paz and Gregorio Martínez-Sánchez
I have not received answers to some of my questions, in particular on the uncertainties, the UV spectra and the unusual coordination via the glucopyranosyl. And I’m still not convinced by the proposed structure of complex III.
There are no major revisions/corrections compared to the first version. Furthermore, there are more questionnings.
I suggest showing the metal-complex HR-MS spectra (with the cluster of isotopic peak) in a supplementary section. Maybe the Table 5 (13C-NMR ) should also be in the supplementary section because it does not give much information. Especially, this does not allow concluding on the coordination sites of the metal. (Example: signal of C’6 have the same shift in peak 2, 3 and 4 for Zn-complexe whereas the authors supposed that the Zn in peak 4 is coordinated to –OH of C’6)
The multiplicity and integration of 1H-NMR should also be given and helpful for the determination of the metal complex structure.
In fact, I’m not sure that the knowledge of the metal complexes structures is important for the second part of the article (the biological activity), because there are no discussion about the relationship between structures and biological activities. The title of the article focuses on the inhibition by these metal mangiferin not on their structure. The authors prove that several complexes are formed with different biological activities. If there are no additional experiments or other evidence for the structure this work could be published in another journal.

Round 3
Reviewer 1 Report
The manuscript is now improved and can be published in Applied sciences.